# Low temperature below 200 °C solution processed tunable flash memory device without tunneling and blocking layer

Sandip Mondal[1,2] & V. Venkataraman[1]

Intrinsic charge trap capacitive non-volatile flash memories take a significant share of the semiconductor electronics market today. It is challenging to create intrinsic traps in the dielectric layer without high temperature processing steps. The main issue is to optimize the leakage current and intrinsic trap density simultaneously. Moreover, conventional memory devices need the support of tunneling and blocking layers since the charge trapping dielectric layer is incapable of preventing the memory leakage. Here we report a tunable flash memory device without tunneling and blocking layer by combining the discovery of high intrinsic charge traps of more than $10^{12}$ cm$^{-2}$, together with low leakage current of less than $10^{-7}$ A cm$^{-2}$ in solution derived, inorganic, spin-coated dielectric films which were heated at 200 °C or below. In addition, the memory storage capacity is tuned systematically upto 96% by controlling the trap density with increasing heating temperature.

[1] Department of Physics, Indian Institute of Science, Bangalore 560012, India. [2] Present address: SanDisk (Western Digital Corporation) India Device Design Center, Bangalore 560103, India. Correspondence and requests for materials should be addressed to S.M. (email: sandip@iisc.ac.in)

Today's semiconductor memory technology is dominated by silicon-oxide-nitride-oxide-silicon (SONOS) non-volatile flash memory which is based on intrinsic charge traps in silicon-rich silicon nitride films deposited by high temperature (equivalent to 780 °C) compatible chemical vapor deposition[1,2]. The intrinsic charge traps in silicon-rich silicon nitride films were first reported in 1967[3] and the first flash memory device incorporating silicon nitride charge storage was demonstrated in 1980's[4]. However the trap density and distribution are difficult to control in such material[5]. Traps can be increased by ion bombardment and plasma-passivation[2], but the leakage current increases. Alternate high-k dielectrics such as $TiO_2$, $HfO_2$, $ZrO_2$, etc. are excellent insulators for transistor applications[6–9], but do not have the intrinsic charge trapping properties as silicon nitride. Although solution processed $HfO_2$ has been used to fabricate SONOS type flash memory[10,11], the devices required the support of additional dielectric layers[12–14], which were deposited by sophisticated ultra-high vacuum techniques with high temperature processing steps to improve the memory leakage. For most dielectrics, precursor solutions with organic solvents result in poor leakage current which can be improved to some extent by high heating temperature[15]. However the high temperature heating process lowers leakage current but reduces trap density. Hence, the main challenge is to simultaneously achieve deep intrinsic charge traps together with very low leakage current at low processing temperatures. There are a few reports on solution processed flash memory by using polymer materials[16–23], but these devices degrade after only few cycles of operation in normal environmental conditions and they are not capable of working at higher temperatures[24–30].

In the last few years, a novel inorganic, completely carbon free, water soluble dielectric aluminum oxide phosphate (ALPO), has been successfully employed as gate dielectric in high performance TFTs that are competitive with a-Si TFTs[31]. Due to its very low leakage current density, it was used recently as tunneling and blocking layer to fabricate fully solution processed two terminal capacitive flash memory devices[32] with CdTe-NP as the charge storage center. Nevertheless a fully spin-coated low temperature processed (below 200 °C) high-performance flash memory device without tunneling and blocking layers has not been reported so far.

Here we report the discovery of ultra high number of intrinsic charge traps (more than $10^{12}$ cm$^{-2}$) in low temperature solution processed inorganic ALPO dielectric. At the same time, the leakage current is sufficiently low (less than $10^{-7}$ A cm$^{-2}$) for flash memory operation without incorporating tunneling or blocking layers. In addition, the number of traps can be varied with heating temperature, which is strongly correlated with the oxygen vacancy concentration in the film. Furthermore, we demonstrate optimized robust high performance fully solution processed inorganic oxide precursor based flash memory devices without tunneling and blocking layer, where the processing temperature does not exceed 200 °C. Our devices outperform other similar memory devices reported earlier[18–23].

## Results

### Charge trapping property at low temperature

A schematic of typical device architecture is depicted in Fig. 1a. ALPO is deposited by spin-coating a completely inorganic, carbon free and aqueous precursor solution on low-doped p-silicon substrate (doping of $4 \times 10^{15}$ cm$^{-3}$). After deposition various substrates are heated at different temperatures including at low temperature such as 200 °C. The inset of Fig. 1a shows a schematic of molecular structure of the low temperature (200 °C) processed ALPO. The thickness of the deposited film is measured by ellipsometry (Supplementary Note 1) and verified by cross-sectional scanning electron microscope (SEM) image (Fig. 1b). Such prepared films are found to be atomically

smooth showing surface roughness to be 0.08 nm[15,33], which is determined by the atomic force microscopy (AFM) with an areal scan over 50 μm × 50 μm (top-panel, Fig. 1c). An aluminum contact is deposited thereafter by thermal evaporation for the purpose of electrical measurement. The deep level charge storage in the deposited ALPO is characterized via capacitance-voltage (CV) measurement. An optical image of device on chip has been shown in the bottom-panel of Fig. 1c.

Intrinsic trap levels in ALPO cause a hysteretic output of the CV traces highlighting the memory-like behavior of the devices (Fig. 1d). Further, the trap density can be varied by heating the samples at different temperatures which alters the hysteresis-window. For example, an as-prepared sample shows a memory window of 12.9 V (left-panel) which corresponds to a trap density ($n$) of $5.65 \times 10^{12}$ cm$^{-2}$, where as, an ALPO-film heated at 200 °C for 1 h shows a hysteresis window of 17.5 V which corresponds to the trap density of $6.37 \times 10^{12}$ cm$^{-2}$. When the film is heated at 600 °C, the trap density drastically reduces to $10^{11}$ cm$^{-2}$ causing a significantly low hysteresis window (equivalent to 2.2 V, right-panel). A similar variation of electronic traps with heating temperature is also observed from the devices which are made with lower thickness ALPO film, however, there is no effect due to change in device dimension (Supplementary Note 3). From the sweep direction of the CV curve it is inferred that gate injection of carriers controls the memory operation. Although conventional flash memory architecture follows channel injection of carriers[2], the interface degrades fast in such devices[34]. Thus, compared to channel injected devices, gate-injected devices show higher endurance which is one of the key requirements of memory operation[35]. Because of such advantages, the quest for efficient gate-injected flash memory devices is ongoing[34–37]. A systematic change in the hysteresis window as a function of heating temperature is presented in Fig. 1e, where, solid squares are the experimental data, shaded region is guide to the eye and the width of the shaded region indicate the standard deviation around their mean vertical position estimated from similar results. Use of higher heating temperature more than 600 °C causes a dramatic reduction in trap density, thus reducing the hysteresis window. A 96% reduction in trap density of $2.19 \times 10^{11}$ cm$^{-2}$) is observed at an heating temperature of 800 °C resulting a memory window of 0.7 V only. In addition, a negligible degradation in CV hysteresis is obtained from as prepared devices even after 5 years of storage in ambient conditions (Supplementary Note 4). This demonstrates that the devices are very stable for long term application, in spite of the water absorbing property of ALPO[38,39].

### Program-disturb (PD) response of memory devices

In order to investigate the robustness of intrinsic charge storage property of ALPO, we demonstrate a series of memory operations on the metal-insulator-semiconductor structured (MIS) two terminal capacitive devices which were heated at 200 °C. The memory characterization focuses upon the flat band voltage shift ($\Delta V_{FB}$) due to the voltage sweep operation (i.e., erasing (E) or programming (P) operation) on the gate terminal of the devices. A device is programmed (P) or erased (E) by sweeping the gate voltage at a slow rate[32]. Sequential P and E cycles set and reset the flatband voltage reversibly and is confirmed from the nearly constant $V_{FB}$ values obtained from multiple E–P cycles (Fig. 2a). Two of the C–V traces corresponding to the 1st and 20th E–P traces are shown in the inset of Fig. 2a. Nearly constant values of $V_{FB}$ in E and P states indicate that E–P cycles have negligible effect on the memory state thus indicating high reliability of the devices.

Frequency and gate bias dependent $\Delta V_{FB}$ changes are shown in Fig. 2b. Flatband voltage window shows insignificant change within the range of 10–1000 kHz frequency of excitation voltage.

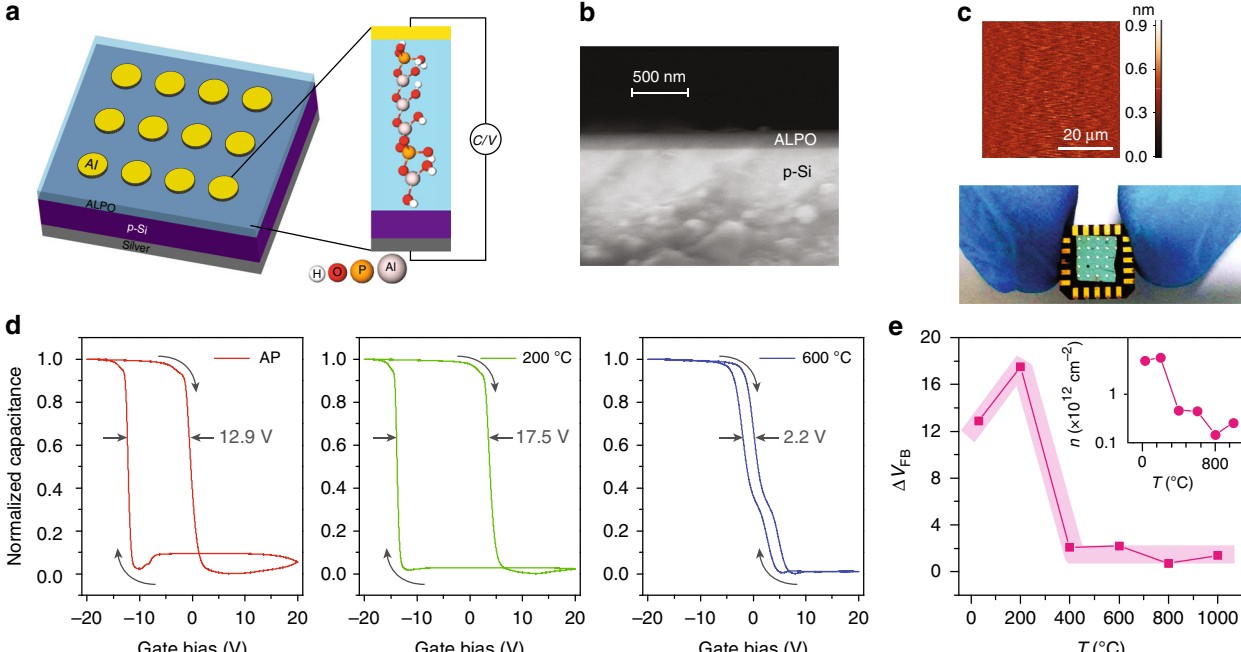

**Fig. 1** Device and deep level charge trapping response. **a** Schematic of device architecture. **b** Cross-sectional scanning electron microscopy (SEM) image of a typical device (thickness of 139 nm). **c** (top-panel) Atomic force topography of the surface of the devices. (bottom-panel) Optical image taken with camera of an array of devices. **d** C–V traces of as prepared (AP), heated at 200 and 600 °C respectively (Supplementary Note 2). The heating was done for 1 h for each sample. C–V was measured with the up-down DC sweep of ±20 V at a rate of 2 V min$^{-1}$ on the gate (Gate Bias) of the metal-insulator-semiconductor (MIS) while imposing a small AC with amplitude and frequency of 100 mV and 100 kHz, respectively. **e** Variation of hysteresis window ($\Delta V_{FB}$) as a function of heating temperature. (inset) variation of trap density as a function of heating temperature

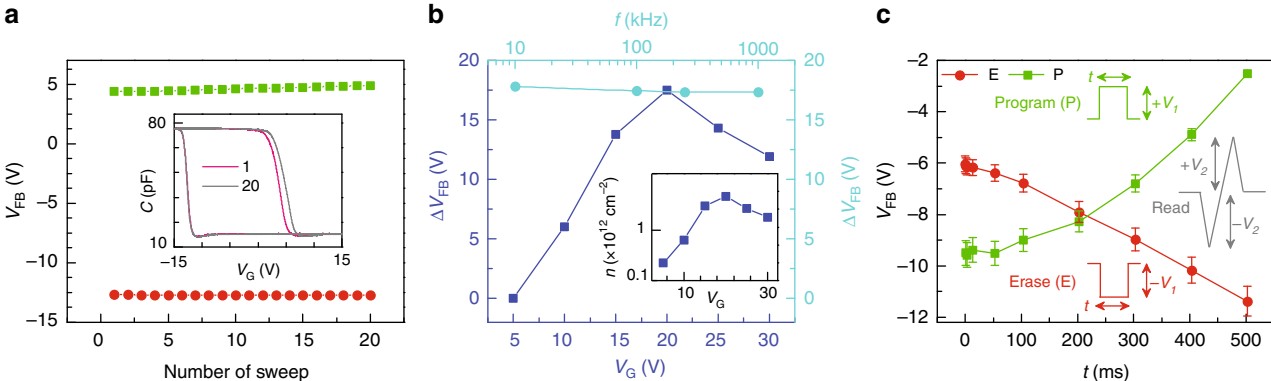

**Fig. 2** Program-disturb (PD) measurement. **a** PD measurement with 20 times C–V sweep on same device. Inset: C–V curve of first and 20th measurement, where $V_{FB}$, $C_{FB}$, and $V_G$ are indicating flatband voltage, flatband capacitance and Gate voltage respectively. **b** Flatband voltage shift ($\Delta V_{FB}$) as a function of gate sweep voltage and frequency of sweep voltage. (Inset) Charge density variation as a function of gate voltage ($V_G$). **c** Programming (P) and erasing (E) operations of the flash memory device as a function of time period (t) of the input pulse. The error bar represents 5% uncertainty in the measured values. Insets indicate shape of the pulses used for program (P), erase (E), and read operations

Such behavior is attributed to the slow trapping de-trapping phenomena which thus indicates that the states can be probed by deploying high speed CV measurement. With different gate bias voltages $\Delta V_{FB}$ increases first and then shows a decrease. Such behavior appears because of the competition between the trap filling and charge losses. Initially (5–20 V) as the sweep voltage is increased, larger electric field causes more charge injection into the traps, hence, the hysteresis width increases. Beyond 20 V, which corresponds to an electric field of 1.2 MV cm$^{-1}$), the high leakage current (Supplementary Note 5) leads to increased charge loss, thus a reduction in hysteresis window is observed (Fig. 2b). A hysteresis window of 15 V is obtained for a sweep voltage range of ±15 V which corresponds to an optimum memory window of 50% of the total sweep-range. The result of 50% window for intrinsic

traps in ALPO exceeds previously reported values[40] for other dielectrics. It is known from literature that the high performance operation of flash memory stack should not have leakage current density more than 10$^{-6}$ A cm$^{-2}$ when operated with an electric field of 1 MV cm$^{-1}$[41]. The ALPO devices show a leakage current density of $4.5 \times 10^{-8}$ A cm$^{-2}$ only at $-1$ MV cm$^{-1}$ electric field. Such low leakage current not only meets the criteria for the application of high-performance flash memory devices, but also outperforms other solution processed inorganic dielectrics namely Al$_2$O$_3$[6], HfO$_2$[7], ZrO$_2$[8], and TiO$_2$[9], which show typical leakage currents of the order of 10$^{-5}$, 10$^{-7}$, 10$^{-2}$, and 10$^{-5}$ A cm$^{-2}$ respectively at 1 MV cm$^{-1}$. This observation for ALPO also ensures that high quality flash memory devices can be fabricated without the need of blocking or tunneling layers.

**Memory performance**. True program (P) and erase (E) operations are realized by application of positive and negative square pulses, respectively, whereas the read operation is performed by using triangular pulses having shorter time periods ($T = 4\,\mu s$). Since high speed voltage sweep does not alter the memory state (Fig. 2b), shorter triangular pulses are expected to probe the E and P states without disturbing them. By varying the width of the square-pulses, various memory-windows ($V_{FB}|_P - V_{FB}|_E$) are obtained and shown in Fig. 2c. While using a single pulse to program or erase, a pulse width of 500 ms can set the memory window to be as large as 9 V. The program/erase speed of these devices is found to be 200 ms, which is significantly faster compared to other solution processed flash memory devices[40,42–45]. During programming and erasing, a maximum charge capturing efficiency of 7.46% is estimated while operating with a gate electric field of 2.37 MV cm$^{-1}$ (Supplementary Note 6). This capturing efficiency is higher than the reported values obtained from other dielectrics[46].

To check for any disturbance of the program/erase state during the read operation, the device is set to E or P states first with a single long square pulse ($T = 0.5$ s, see Fig. 2c) and then multiple high-frequency triangular ($T = 4\,\mu s$) pulses were applied representing multiple read operations. Figure 3a shows the statistical distribution of $V_{FB}$ values obtained from such multiple read operations for both the E and P states. Narrow distributions of $V_{FB}$ indicate low read-disturb for both the states. Inset of Fig. 3a shows typical C–V traces for E, P, and read operations. Here, two red and two green lines indicate back-and-forth sweep of the voltage ($V_G$) using the high speed triangular read pulse probing the P and E states, respectively.

Erase-program operation over 10 k cycles shows no degradation in memory window thus demonstrates high endurance (Fig. 3b). In fact, with the increasing operation cycle memory window increases which may be attributed to the generation of additional trap states because of electrical stress. Data retention was tested by programming a fresh device at room temperature with pulses of amplitude 33 V and duration 500 ms. The device shows almost no change in memory window within $10^4$ s which was the limit of experimental time scale. This is the highest reported retention time for any memory device without additional tunneling and blocking layers. Even after 10 k P/E cycles devices show a degradation of only 9% memory window after $10^4$ s (Fig. 3d). Temperature dependent data retention was also tested and presented in Fig. 2e. After $10^4$ s of operation a memory window loss of 25 and 50% are observed at 60 and 80 °C respectively. The reliability is also tested on the device made with lower thickness of ALPO film (91 nm) which shows equivalent performance as obtained from the other devices (Supplementary Note 7).

To examine the scalability of these low temperatures processed memory devices, we prepared and characterized more than 40 devices. Statistical distribution of memory window for all these devices are shown in Fig. 3f. Eighty percent of devices lie within the expected memory window (3–4 V) when programmed with ±33 V pulse height and 300 ms duration. The remaining 20% devices show a memory window of less than 3 V. These preliminary results show sufficiently high yield for large scale production needed for practical applications.

## Discussion

Microscopic origin of memory operation is understood from the temperature controlled trap density variation which is confirmed from photo-emission spectroscopy (XPS). Multiple samples were prepared at different heating temperatures and studied with XPS. The shape of the oxygen peak changes systematically with respect to heating temperature (Fig. 4a). De-convolution of XPS signal

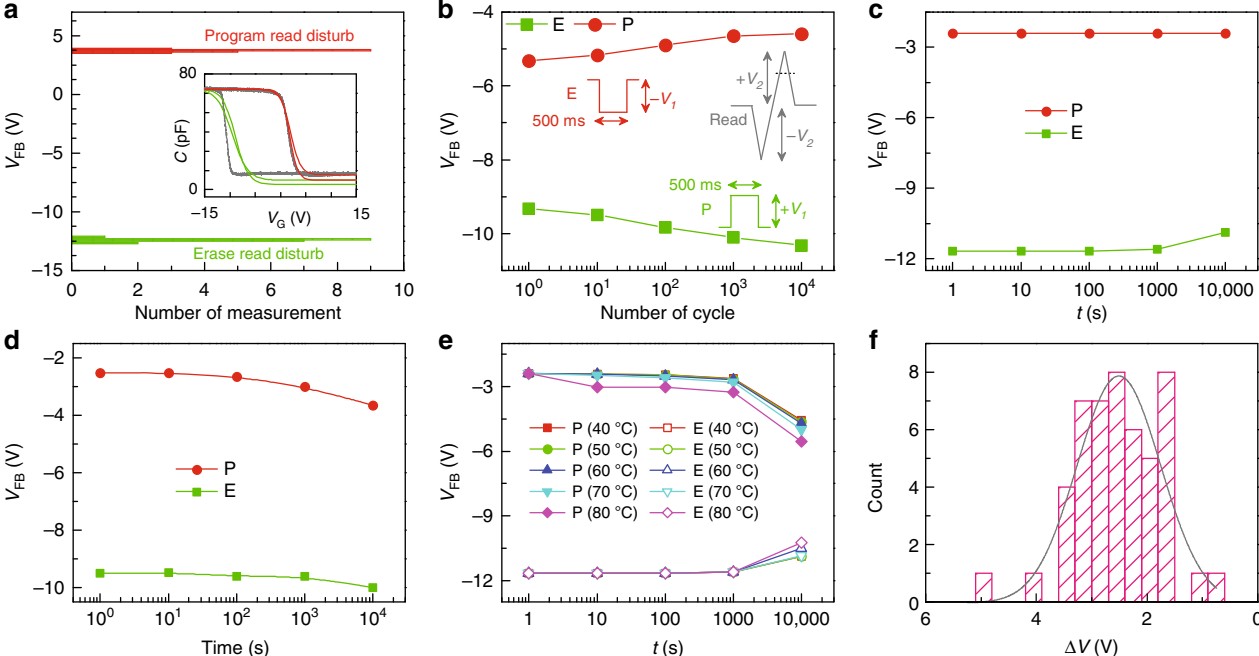

**Fig. 3** Robust flash memory operation. **a** Program-disturb (PD) verification with high-speed C–V measurement system. $V_{FB}$ indicates flatband voltage. (Inset) capacitance (C)–voltage ($V_G$) traces from low speed (gray-traces) and high speed (red and green traces) measurements. Red colored lines indicate back-and-forth sweeps at high speed (time period ($T$) equals to $4\,\mu s$) while the sample is in programmed state. Green colored lines indicate back-and-forth sweeps at high speed (time period ($T$) equals to $4\,\mu s$) while the sample is in erased state. **b** Endurance characteristics of the device. (Inset) Shape of the pulses used in program (P), erase (E), and read operations. **c** Retention characteristics of the flash memory device. Programming of the state was done with a square pulse of height ±33 V and having a duration of 500 ms. **d** Retention characteristics of the flash memory device after endurance test of 10 k cycles. **e** P/E test of the flash memory devices at various temperatures. Lines are the guide to eye. **f** The statistical distribution of flash memory window for 47 devices. For this test applied P/E voltage is ±33 V for 300 ms

into respective peaks of oxygen vacancies (M-O$_{vac}$), metal hydroxide (M-OH), and lattice oxygen (M-O) helps understanding the contributions of respective states (Fig. 4b–d). The ALPO film which was heated at 200 °C has similar ratio (atomic %) of aluminum (Al), oxygen (O), and phosphorous (P) as that of the mother solution (Supplementary Note 8). Due to addition of HCl in the ALPO precursor, a small percentage of chlorine is observed (Supplementary Note 9) in low temperature processed ALPO film. It is observed that M-O$_{vac}$ intensity sharply decreases with increase in heating where temperature reaches above 200 °C (Supplementary Note 10). As the heating temperature increases, hydrogen is lost as water vapor and the film becomes denser. The M-O peaks increase with higher heating temperature (Fig. 4e, f). This is consistent with the reduction in the number of oxygen vacancies and consequently lower trap density in samples heated at higher temperature. A quantitative comparison between the peak intensity and the trap density of the heated film reveals a strong correlation between the oxygen vacancies and trap densities (Supplementary Note 10), thus indicating that the oxygen vacancies are the responsible entities for memory states.

In conclusion we have shown fabrication and high quality performance of low temperature processed (less than 200 °C) inorganic flash memory devices which do not require tunneling or blocking layers. Simple sample fabrication technique involving spin-coating of solution is another advantage. Narrow distribution of memory window obtained from more than 40 samples indicates the scalability of the fabrication method. In spite of having no tunneling and blocking layers, these devices show extremely low leakage current which is one of the key features of memory operation, and demonstrate high endurance, high retention, thus, outperforming other memory devices reported so far (Table 1). Temperature dependent control on trap density also

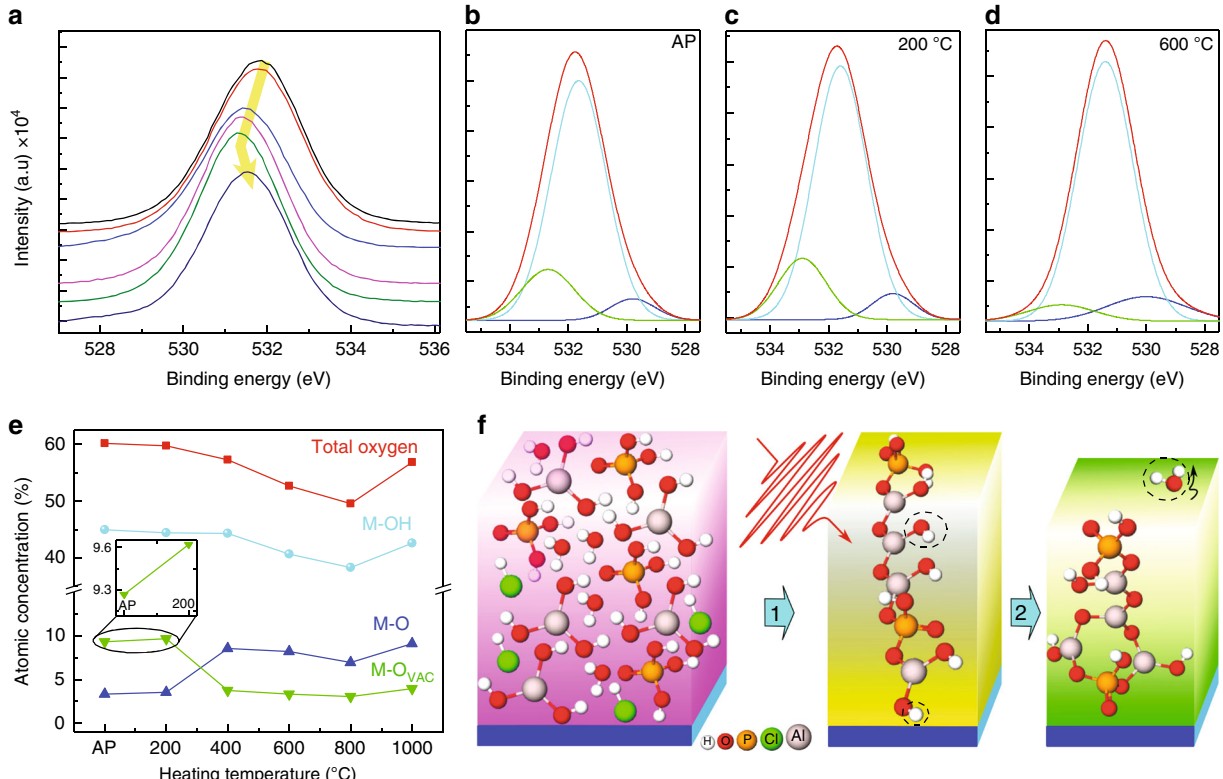

**Fig. 4** Temperature dependent changes in trap states. **a** Core level O 1s spectra from different temperature heated film. Thick arrow indicate movement of the peaks obtained from samples heated with different temperatures. **b**–**d** Core level oxygen peaks from as-prepared, 200 and 400 °C heated ALPO films, respectively. The oxygen peak (red) in each case is de-convoluted into three components corresponding to oxygen vacancies (M-O$_{vac}$) is green, lattice oxygen (M-O) is blue, and metal hydroxide (M-OH) is cyan. **e** Atomic composition ratios of oxygen in ALPO thin films as a function of temperature. (inset) increment of traps O$_{VAC}$ at 200 °C with respect to AP ALPO film. **f** Schematics showing condensation mechanism of oxide precursors by heating. The first block denotes the individual molecules, second and third are low and high temperature processed ALPO films, respectively. All molecular models were constructed using MolView (http://molview.org/). Escape of water molecules (black circle) shown in third block leads to the decrease in the trap states as well as the decrease of the thickness of the film

### Table 1 Memory technology comparison

| Device fab. technique | Structure | P/E vol (V) | P/E time | Memory window | Ref. |
|---|---|---|---|---|---|
| ALD and RF sputtering | p-Si/SiO$_2$/Ge-NC/TaZrO$_x$/Al$_2$O$_3$/Al | 10 V | 60 s | 3.8 V | 40 |
| PECVD and thermal | Si/SiO$_2$/Al@Al$_2$O$_3$-NP/SiO$_2$/Al | ±15 V | 5 s/5 s | 2.5 V | 42 |
| PVD and e-beam | p-Si/Al$_2$O$_3$/Pt-NP/Al$_2$O$_3$/Ti | ±7 V | 5 s/5 s | 4.3 V | 43 |
| RF sputtering | p-Si/Al$_2$O$_3$/GeNC1L/Al$_2$O$_3$/Al | 5 V/−5 V | 3 s/3 s | 2.3 V | 44 |
| PECVD and sputtering | p-Si/SiO$_2$/DyTi$_x$O$_y$/Al$_2$O$_3$/Al | 7 V/−10 V | 1 s/10 s | 4 V | 45 |
| Solution processed | p-Si/ALPO/Al | 30 V/−30 V | 500 ms/500 ms | 9 V | This work |

helps optimizing the memory window. ALPO based devices paves the way for designing a new class of scalable two terminal flash memory devices for practical applications.

## Methods

**Material growth**. Precursor solution of aluminum oxide phosphate, ALPO [$Al_2O_{3-3X}$ $(PO_4)_{2X}$] in water (18 MΩ cm) was prepared with $Al(OH)_3$ (99%, Alfa Aesar, USA) in 2 mole equivalents of HCl (AR Grade, Thermo Fisher Scientific, USA) and an appropriate amount of $H_3PO_4$ (ExcelaR Grade, Thermo Fisher Scientific, USA) was added to obtain P/Al = 0.5 with concentration of 0.5 M. The solution was stirred under heat of 90 °C in a water bath for 24 h.

**Device fabrication**. Two terminal MIS structures were fabricated on piranha cleaned [$H_2SO_4$:$H_2O_2$ = 3:1] lightly doped silicon substrate (p–Si). The ALPO solution was spin-coated at 3000 rpm for 30 s to form the gate dielectric. The films were then heated at 150 °C for 1 min. This process was repeated for 2–3 times to achieve the desired thickness. The sample was exposed to oxygen plasma for 5–10 min at 0.5 mbar pressure before deposition of each layer. Such fabrication process may help generation of stable traps in ALPO below 200 °C. Typically, ALPO-based devices are treated at high temperature (more than 350 °C) to achieve better dielectric performance, thus, most of the reported devices do not show any memory effect[31]. For control devices the film was further heated at 800 °C for 1 h in ambient to achieve the trap free oxide. A 200 nm aluminum gate was deposited by thermal evaporation at $10^{-6}$ mbar pressure. Before deposition of the top aluminum gate, the ALPO films were heated at different temperature in a preheated furnace for 1 h. All devices were stored under ambient conditions and no degradation of CV curves was found even after 5 years.

**Characterization**. ALPO precursor solutions prepared with different concentrations and P/Al ratios were optically characterized using a UV–Visible spectrophotometer. They are all transparent in visible wavelength range with the main absorption peak in the ultra-violet range at 235 nm. After spin coating and heating, the film shows a refractive index 1.5 (Supplementary Note 1) with negligible absorption at 550 nm as measured by ellipsometry (M-2000, J.A. Woollam Co. Inc., USA). The thickness of ALPO film on silicon substrate extracted from ellipsometry is 139 nm and verified with cross-sectional SEM (Ultra 55, Carl Zeiss). The surface of the ALPO film is atomically smooth with a roughness of 0.08 nm as measured by AFM (ND-MDT, Russia). The CV curve was measured with HIOKI 3532 LCR meter and Keithley 2400 source meter. Agilent Device Analyzer B1500A was used to measure the IV characteristics in ambient environment. The leakage current and CV measurements were performed on more than 50 devices. The entire measurement was done in continuous mode of the instrument. The high speed CV measurement was performed with home made CV measurement system[32]. The XPS measurements were carried out by AXIS 165 with Al Kα radiation (9 mA, 13 keV, and 1486.6 eV) in ultra-high vacuum. The XPS spectra were calibrated with C 1s peak (284.6 eV).

## Data availability

The authors declare that all data supporting the findings of this study are included in the main manuscript file or Supplementary Information or are available from the corresponding author upon request.

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

## Acknowledgements

This work was supported by the grant (MITO−084) of Center for Nano Science and Engineering (CeNSE), Indian Institute of Science, Bengaluru, India. S.M. would like to thank Dr. Kallol Roy for useful discussions during manuscript preparation. S.M. also thanks Mr. Chandan and Dr. Arvind Kumar for useful discussion. S.M. also thanks the Rajiv Gandhi National Fellowship of the University Grants Commission (UGC) for fellowship support.

## Author contributions

S.M. and V.V. conceived the project. S.M. designed and performed the experiments, completed characterization of the fully solution processed flash memory device. S.M. analyzed results and wrote the paper.

## Additional information

**Competing interests:** The authors declare no competing interests.

