## [Peer Review File · Nature Communications]

Reviewers' comments:

Reviewer #1 (Remarks to the Author):

S. Mondal et al developed a low temperature solution-processible flash memory device using aluminum oxide phosphate (ALPO) gate dielectric of high number of charge traps ($\sim 10^{12} \text{ cm}^{-2}$). The discovery of the dielectric ALPO as a gate-injected charge trapping layer enabled the simplification of the device configuration by avoiding incorporation of conventional tunneling and blocking layers. Moreover, the device shows high performance device operation such as long retention time ($\sim 10^4 \text{ s}$), high endurance ($> 10^6$), and low leakage current level ($< 10^{-7} \text{ A cm}^{-2}$) even without tunneling or blocking layers. The authors demonstrated the correlation of the number of traps with oxygen vacancy concentration as a function of processing temperature mainly by XPS measurement and claimed that the charge trap density became high at a certain temperature range of approximately 200 °C. It is interesting to see a novel charge trapping layer potentially suitable for a flash memory device even without sophisticated interlayers of blocking and tunneling ones. The work seems, however, in preliminary stage which requires substantial study to elucidate the details of the charge trapping mechanism and the origin of the temperature dependent charge trapping performance. More importantly, the reviewer hardly sees the novelty of the work in terms of materials design and fabrication process except their finding of a proper thermal treatment condition at 200 °C. This indicates at the same time that the memory performance is process-temperature dependent, possibly resulting in a memory unstable at temperature. The rather poor retention results at 60 °C shown in the manuscript support the reviewer's argument. In addition, as the authors briefly mentioned, non-volatile memory performance should be demonstrated in the levels of arrays as well as lateral injection type transistors to broaden the usage of the trapping layer. In addition to XPS results, further systematic investigation should be performed with a variety of characterization tools to convince the authors' claims. The known insulating properties of ALPO and its charge trapping properties shown in the current work should be more clearly controlled and explicitly distinguished with more experimental results. As the authors are aware, there are numerous non-volatile memory works associated with materials contamination by oxygen, water, residual solvent and so on due to incomplete materials processes. To clearly differentiate the work with others, the manuscript should be returned to the authors to address these issues carefully in addition to more specific questions shown below.

1. In Fig 1d, the hysteresis window increased from 12.9 V to 17.5 V after 200 °C thermal irradiation, meaning that the charge trap density (i.e., oxygen vacancy as the author maintained in Fig 4) significantly increased. However, in Fig 4e, M-Ovac (oxygen vacancy) didn't increase after 200 °C thermal irradiation. Please explain this difference.

2. What's the physical origin of low leakage current of ALPO compared with dielectrics like Al_2O_3 , HfO_2 , ZrO_2 , TiO_2 ? Al_2O_3 or HfO_2 is known as good insulating materials, how can ALPO show lower leakage current than them? Please explain this.

3. The authors maintain that the switching speed of 200 ns of the ALPO device is much faster than the referred reports ([3], [5]-[7], [12]). However, the papers that the authors referred include organic materials or transistor-type devices. How faster is it compared with inorganic flash memory devices with MIS capacitive type? In addition, the author should refer more recently reported papers.

4. The thermal stability is not good, as the retention degrade by 25 % only above 60 °C. Is it intrinsic problems of ALPO? How can it be improved?

5. Scalability of a non-volatile memory is also important. Device dimension dependent memory performance should be addressed.

6. Low processing temperature is beneficial, in particular, when employed to a polymer substrate. It would be good to see non-volatile flash memory with mechanical flexibility or on a plastic substrate.

Reviewer #2 (Remarks to the Author):

The paper investigates a novel solution-processed tunable Flash memory based on the adoption of a sol-gel dielectric ALPO.

The application of this novel dielectric to the Flash memory domain is novel, despite there are points, not addressed in the current version of the manuscript, that must be explored to understand the full potential of this device concept from an application point of view, e.g. what happens with reducing the gate voltage and the dielectric thickness on leakage current and reliability (retention and endurance)? Can the spin coating fabrication techniques fit into the 3D process, which is the current mainstream for Flash memories? What about temperature dependence?

I think the above points need to be covered to provide further evidence of the potential of the device concept, and to influence thinking in the field.

On the other side, the paper is well written and organized. Conclusions are supported by a good statistical analysis of data, despite also the variation of the dielectric thickness (as reported below) needs to be included to provide evidence that the device full operation is maintained for lower thickness and gate voltages. In case, this is not verified, the application areas of the paper are much narrower and probably restricted to embedded/wearable fields.

Reviewer #3 (Remarks to the Author):

The authors provide a concise description of C-V hysteresis in metal-insulator-semiconductor devices fabricated with the solution-processed insulator AIPO. They claim high trap densities, low leakage currents, and reproducible behavior offer the potential for broader impacts in memory applications. I support publication of the work after the authors address a few items:

"thermally irradiated" can be changed to "heated"

The solution method is not traditional "sol-gel". A homogeneous salt solution converts to an amorphous, continuous film during spin coating; a conventional sol does not form.

Is all the chloride expelled from the film at 200 C?

The term "trap states" is probably a better descriptor than "defect states", since the films are amorphous.

I don't understand the origins of "Ovac" (oxygen vacancy) in the XPS studies. The films are amorphous and the temperatures are too low to produce any measurable substoichiometric oxygen concentrations; oxide, hydroxide, and phosphate concentrations will be balanced stoichiometrically by H⁺, Al³⁺, and P⁵⁺.

The films have high -OH content and attendant porosity relative to a high-density oxide film, i.e., the films formed by heating at 600 C and higher. Pores lined with -OH could be primary trap states and contributor to hysteresis.

The purities of the reagents should be reported in the experimental section. Impurities could contribute to trap-state concentrations.

AIPO deposited from solution is very sensitive to environmental humidity. The films reversibly absorb and desorb water as humidity and temperature change. The water content could affect the long-term reproducibility and stability of the proposed devices. The authors may wish to consider the following publications, which may provide ideas to affect device performance.

Solid State Sciences 61, 106-110 (2016).

ACS Sustainable Chemistry & Engineering 3, 1081-1085 (2015)

To,
The Reviewers
Nature Communications

21th January 2019

We wish to thank the Reviewers for spending valuable time to read the manuscript (NCOMMS-18-34803-T), and making pertinent and insightful comments. We found every comment extremely helpful, and incorporating these improved the quality of the manuscript as well as our understanding on some of the fundamental aspect of the materials and devices. In order to address all the comments of the referees, we carried out new measurements with new sets of devices, thus, it took us some time to respond to the comments. We believe that we have been able to address all the concerns of the reviewers in the modified version of the manuscript. The key changes in the modified manuscript are mentioned below

1. A set of new devices have been fabricated with lower thickness of ALPO (~ 47nm as well as 91 nm) and annealed at different temperature. The output of such devices has been demonstrated in the Section III, Supplementary Information. It is also demonstrated that the thickness normalized response appears similar in all cases indicating operational scheme at lower dielectric thickness.

2. The stability of the devices in open lab environment has been proven by comparing two sets of results recorded after a span of ~ 5-years (Section IV, Supplementary Information).

3. The reliability of the devices with lower thickness of dielectric film has been studied which shows comparable performance with respect to the devices made with thicker ALPO film (Section VII, Supplementary Information).

4. A set of XPS measurement has been performed on ALPO film annealed at different temperature and demonstrate the chlorine (Cl) concentration which is shown in Section IX, Supplementary information.

5. We have compared the recent technology and put up a Technology Comparison Table-I in the Main manuscript.

6. The reference list and figure captions are also updated.

Point-by-point reply to the comments of the Reviewer is appended below. We hope that the modified version of the manuscript will be acceptable for publication in Nature Communications journal.

Yours sincerely,

Sandip Mondal
V Venkataraman

SanDisk | a Western Digital brand India Device Design Centre
Department of Physics, Indian Institute of Science, Bangalore-560012

Reviewer #1 (Remarks to the Author):

General Comment:

S. Mondal et al developed a low temperature solution-processible flash memory device using aluminum oxide phosphate (ALPO) gate dielectric of high number of charge traps ($\sim 10^{12}$ cm⁻²). The discovery of the dielectric ALPO as a gate-injected charge trapping layer enabled the simplification of the device configuration by avoiding incorporation of conventional tunneling and blocking layers. Moreover, the device shows high performance device operation such as long retention time ($\sim 10^4$ s), high endurance (>10 k), and low leakage current level ($<10^{-7}$ A cm⁻²) even without tunneling or blocking layers. The authors demonstrated the correlation of the number of traps with oxygen vacancy concentration as a function of processing temperature mainly by XPS measurement and claimed that the charge trap density became high at a certain temperature range of approximately 200°C. It is interesting to see a novel charge trapping layer potentially suitable for a flash memory device even without sophisticated interlayers of blocking and tunneling ones. The work seems, however, in preliminary stage which requires substantial study to elucidate the details of the charge trapping mechanism and the origin of the temperature dependent charge trapping performance. More importantly, the reviewer hardly sees the novelty of the work in terms of materials design and fabrication process except their finding of a proper thermal treatment condition at 200°C. This indicates at the same time that the memory performance is process-temperature dependent, possibly resulting in a memory unstable at temperature. The rather poor retention results at 60°C shown in the manuscript support the reviewer's argument. In addition, as the authors briefly mentioned, non-volatile memory performance should be demonstrated in the levels of arrays as well as lateral injection type transistors to broaden the usage of the trapping layer. In addition to XPS results, further systematic investigation should be performed with a variety of characterization tools to convince the authors' claims. The known insulating properties of ALPO and its charge trapping properties shown in the current work should be more clearly controlled and explicitly distinguished with more experimental results. As the authors are aware, there are numerous non-volatile memory works associated with materials contamination by oxygen, water, residual solvent and so on due to incomplete materials processes. To clearly differentiate the work with others, the manuscript should be returned to the authors to address these issues carefully in addition to more specific questions shown below.

Reply:

We thank the Reviewer for these comments. We report a striking persistent memory with temperature controllable memory effect obtained from a low-temperature treated solution processed dielectric, which have importance both in fundamental and technological grounds. In total there are THREE novel and fundamental aspects of this work.

(1) The memory/trap density is 'tuneable' and can be controlled with simple thermal irradiation technique. The processing temperature can be as low as 200°C. The controllable traps in fully solution processed, spin-coated dielectric itself is the very first report of its kind. Such controllable charge storage remains a challenge even in the well-established Si₃N₄ technology.

(2) From the perspective of memory device, this material does not need the help of supporting dielectrics layers (tunnelling or blocking layers) to prevent the memory leakage. In spite of having no tunnelling and blocking layers, such devices show near perfect memory properties which is rarely seen in solid state electronic devices. Thus, makes these devices attractive for non-volatile flash memory applications.

(3) A strong correlation between the controllable memory-state and material property (trap density) in sol-gel dielectric has been demonstrated.

Indeed, a table comparing the technological impact on memory properties is also added. Following the Referee's comments, we have overhauled the focus of the manuscript to emphasize these aspects. We have performed a series of XPS survey measurement on different temperature processed ALPO film and explained the result more elaborately. We wish to have your opinion on the manuscript in its modified form.

Question 1A:

In Fig 1d, the hysteresis window increased from 12.9 V to 17.5 V after 200 °C thermal irradiation, meaning that the charge traps density (i.e., oxygen vacancy as the author maintained in Fig 4) significantly increased. However, in Fig 4e, M-Ovac (oxygen vacancy) didn't increase after 200°C thermal irradiation. Please explain this difference.

Reply:

Oxygen vacancy increases from annealing temperature of AP to 200°C as demonstrated in Fig 1d as well as in Fig 4. For more clarity, we have added an inset in Fig 4e. The quantitative analysis of the oxygen vacancies are reported in supplementary (Section X, Supplementary Information).

Question 1B:

What's the physical origin of low leakage current of ALPO compared with dielectrics like Al₂O₃, HfO₂, ZrO₂, TiO₂? Al₂O₃ or HfO₂ is known as good insulating materials, how can ALPO show lower leakage current than them? Please explain this.

Reply:

Typically, Al₂O₃, HfO₂, ZrO₂, TiO₂ etc. are grown in alcohol/organic precursor where presence of hydro-carbon contaminant may lead to higher leakage current (Ref: *IEEE Trans on Elec Dev*, vol. 44, no. 11, 1997, DOI: 10.1109/16.641371). On other hand ALPO is grown from water soluble precursor, hence, completely free from hydro-carbon contaminants leading lower leakage current. In addition, the atomically smooth nature of ALPO film is expected to demonstrate lower leakage current [Appl Surf Sci 256 (2010) 6667–6672, DOI:10.1016/j.apsusc.2010.04.067 and Phys. Rev. B 60, 9157 (1999), DOI: 10.1103/PhysRevB.60.9157).

Question 1C:

The authors maintain that the switching speed of 200 ms of the ALPO device is much faster than the referred reports ([3], [5]-[7], [12]). However, the papers that the authors referred include organic materials or transistor-type devices. How faster is it compared with inorganic flash memory devices with MIS capacitive type? In addition, the author should refer more recently reported papers.

Reply:

We agree with the Reviewer, and added a table (Table-1) in the main manuscript, which compares various parameters (film deposition techniques, programming speed, memory window etc) of our devices with those from existing MIS capacitive type memory technology based on inorganic materials fabricated by using sophisticated ultra-high vacuum techniques. Also the list references have been revised with more recent one.

Although we are comparing our devices with the existing technology, however, it is to be noted that our devices are made without tunnelling and blocking layers which is a significant challenge in current technology. Indeed, there are no such fully solution processed memory device available which has been made by utilising the property of intrinsic deep level traps in dielectrics.

Question 1D:

The thermal stability is not good, as the retention degrade by 25% only above 60°C. Is it intrinsic problems of ALPO? How can it be improved?

Reply:

We agree with the Reviewer. The memory leakage at higher temperatures may come from the shallow energy levels of the traps, thus, gives us the scope for future study. One of the solutions to this problem would be to generate deep level traps by annealing these devices in oxygen rich environment (similar is seen when HfO₂ is annealed in O₂ environment, Reference: *Applied Physics Letters* 105, 172902 (2014), DOI: 10.1063/1.4900745).

Question 1E:

Scalability of a non-volatile memory is also important. Device dimension dependent memory performance should be addressed.

Reply:

We agree with the reviewer. We have now expanded our study to include the performance of the devices with lower thickness. A set of new devices have been fabricated with lower thickness of ALPO (~ 47nm as well as 91 nm) and annealed at different temperature. The memory characteristics of these new devices is now included in Section III, Supplementary Information. It is also demonstrated that the thickness normalized response appears similar in all cases indicating good scaling behaviour.

Question 1F:

Low processing temperature is beneficial, in particular, when employed to a polymer substrate. It would be good to see non-volatile flash memory with mechanical flexibility or on a plastic substrate.

Reply:

We thank the Reviewer for this comment. It is true that the potential application of this invention of deep level intrinsic traps in solution processed ALPO is useful for flexible electronics application. The focus of this manuscript is to demonstrate the fundamental invention of tunable intrinsic traps in sol-gel dielectrics. The fabrication of memory device on flexible substrate may be the scope for future work.

Reviewer # 2 (Remarks to the Author):

Question 2A:

The paper investigates a novel solution-processed tunable Flash memory based on the adoption of a sol-gel dielectric ALPO. The application of this novel dielectric to the Flash memory domain is novel, despite there are points, not addressed in the current version of the manuscript, that must be explored to understand the full potential of this device concept from an application point of view, e.g. what happens with reducing the gate voltage and the dielectric thickness on leakage current and reliability (retention and endurance)?

Reply:

Thank you for such query. We have supported the results with the gate voltage dependent data which is available in the supplementary (Section II, Supplementary Information).

Indeed, we have fabricated memory devices with lower thickness (~47nm as well as 91nm) of ALPO film (Section III, Supplementary Information) and demonstrated a significantly large hysteresis window. The thickness normalized flatband voltage appears similar in all cases indicating operational scheme at lower dielectric thickness. Typical leakage current values for 91nm devices are measured to be $\sim 7.8 \times 10^{-8}$ A.cm⁻² at 2MV/cm⁻¹ (Sandip Mondal et al., *Appl. Phys. Lett.* 111, 041602 (2017), DOI: 10.1063/1.4995982)

The data retention and endurance test of 91nm thick ALPO devices has been performed. We do not see any degradation in retention and endurance because of the lower dielectric thickness (Section VII, Supplementary Information).

Question 2B:

Can the spin coating fabrication techniques fit into the 3D process, which is the current mainstream for Flash memories? What about temperature dependence?

I think the above points need to be covered to provide further evidence of the potential of the device concept, and to influence thinking in the field.

Reply:

The spin-coating fabrication can be implemented into the 3D NAND memory technology. However, the focus of this manuscript is to demonstrate the discovery of deep level intrinsic traps in solution processed dielectric which is useful for low cost memory technology application. Fabrication of 3D NAND memory cell through the solution route is the focus of the futuristic interest.

We have thoroughly studied the temperature dependent tunability effect of the memory technology which is shown in the Fig. 1 d-e.

Question 2C:

On the other side, the paper is well written and organized. Conclusions are supported by a good statistical analysis of data, despite also the variation of the dielectric thickness (as reported below) needs to be included to provide evidence that the device full operation is maintained for lower thickness and gate voltages. In case, this is not verified, the application areas of the paper are much narrower and probably restricted to embedded/wearable fields.

Reply:

We thank the Reviewer for this comment. We have included charge trapping phenomena from a new set of devices having different ALPO thickness in now included in the supplementary information (Fig. S3b of Section III, Supplementary Information). Such study confirms that the change in ALPO thickness does not affect the charge trapping property.

Reviewer # 3 (Remarks to the Author):

General Comment:

The authors provide a concise description of C-V hysteresis in metal-insulator-semiconductor devices fabricated with the solution-processed insulator AlPO. They claim high trap densities, low leakage currents, and reproducible behaviour offer the potential for broader impacts in memory applications. I support publication of the work after the authors address a few items:

Reply:

We thank the Reviewer for acknowledging our work. Following the comments we have modified the manuscript.

Question 3A:

"thermally irradiated" can be changed to "heated"

Reply:

We have changed this phrase. Respective changes have been highlighted.

Question 3B:

The solution method is not traditional "sol-gel". A homogeneous salt solution converts to an amorphous, continuous film during spin coating; a conventional sol does not form.

Reply:

We agree with the Reviewer. We have removed the "sol-gel" terminology from manuscript and indicated the fabrication process to be a "solution processed" at appropriate places. Respective changes have been highlighted in the manuscript.

Question 3C:

Is all the chloride expelled from the film at 200 C?

Reply:

A very low amount of chlorine is present in the ALPO film which is heated at AP and 200°C. Hence we have added this information in the main manuscript with supporting data in supplementary (Section IX, Supplementary Information). We have extended the manuscript with the information such as,

"In addition, a small percentage of chlorine is present (Section IX, Supplementary Information) in low temperature processed ALPO film due to addition of HCl in the ALPO precursor."

Question 3D:

The term "trap states" is probably a better descriptor than "defect states", since the films are amorphous.

Reply:

We agree with the Reviewer the reviewer comment hence modified the manuscript accordingly with highlighter.

Question 3E:

I don't understand the origins of "Ovac" (oxygen vacancy) in the XPS studies. The films are amorphous and the temperatures are too low to produce any measureable substoichiometric oxygen concentrations; oxide, hydroxide, and phosphate concentrations will be balanced stoichiometrically by H⁺, Al³⁺, and P⁵⁺.

Reply:

We thank the reviewer for this observation. Although our films are annealed at low temperatures, the XPS data clearly shows the presence of oxygen vacancies in the ALPO film. The exact mechanism of vacancy formation in ALPO is not yet known. Nevertheless oxygen vacancies are typically observed in solution processed metal-oxide films annealed at low temperature as reported by other groups [Ref 1, 2, 3].

References:

- 1) *Nature Materials* **10**, 45–50 (2011), DOI: 10.1038/nmat2914
- 2) *Nature Materials* **10**, 382–388 (2011), DOI: 10.1038/nmat3011
- 3) *Nature* **489**, 128–132 (06 September 2012), DOI: 10.1038/nature11434

Question 3F:

The films have high -OH content and attendant porosity relative to a high-density oxide film, i.e., the films formed by heating at 600 C and higher. Pores lined with -OH could be primary trap states and contributor to hysteresis.

Reply:

We thank the reviewer for this comment. We are not ruling out the other possibility which may contribute to trap states in the low temperature processed ALPO. However, in case of OH, it is observed from our data that -OH content remain similar for AP, 200°C and 400°C (Fig. 4e), whereas, the flatband voltage decreases drastically at lower 400°C (Fig. 1e). On the other hand, the change in hysteresis window completely follows the oxygen vacancy concentration variation in different temperature processed ALPO film. Hence, we attribute the trapping phenomena to the availability of oxygen vacancies.

Question 3G:

The purities of the reagents should be reported in the experimental section. Impurities could contribute to trap-state concentrations.

Reply:

We thank the reviewer for this comment. We added the property of the chemicals which has been used to grow the ALPO solution. We have highlighted the information in the experimental section.

Question 3H:

ALPO deposited from solution is very sensitive to environmental humidity. The films reversibly absorb and desorb water as humidity and temperature change. The water content could affect the long-term reproducibility and stability of the proposed devices.

Reply:

We kept these devices in open lab environment for more than 5 years as shown in Section IV, Supplementary Information. There is negligible degradation observed in hysteresis window, which proves that ALPO film offers stable and reproducible memory devices for long term application.

Question 3I:

The authors may wish to consider the following publications, which may provide ideas to affect device performance.

Solid State Sciences **61**, 106-110 (2016).

ACS Sustainable Chemistry & Engineering **3**, 1081-1085 (2015)

Reply:

We are thankful to the reviewer for the suggesting these articles which appears very relevant as they highlight in-depth study of ALPO characteristics. We have included these references at appropriate places.

Reviewers' comments:

Reviewer #1 (Remarks to the Author):

The authors addressed most of the concerns the reviewer raised with extra experiments and analysis. The revised manuscript may be suitable for the publication with additional clarification. One thing still unclear to the reviewer is what the breakthrough of the work is in terms of materials development. ALPO is known as an insulating material and used as a dielectric for low leakage current. The reviewer understands that the trapped charges are controlled by the temperature and thus utilized for a novel non-volatile memory, but the question is whether others simply overlooked this simple temperature-dependent charge control or not. Otherwise, the authors specially treated ALPO for the purpose. This part is not clear. The scalability the reviewer mentioned was the area scalability rather than thickness one. As the authors are aware, the areal dimension should be as small as possible for the high density memory arrays. In fact, some of non-volatile memories lose their memory properties when the capacitor area is reduced to a few tens of nanometers. The reviewer supports the publication after clarification of these issues.

Reviewer #2 (Remarks to the Author):

No other comments from my side here.

To,
The Reviewers
Nature Communications

19th February 2019

We wish to thank the reviewers for spending valuable time to read the manuscript (NCOMMS-18-34803A), and making pertinent and insightful comments. We also thank reviewer-1 for the comments which helped improving the quality of the manuscript further. In order to address the comment of the reviewer-1, we carried out new measurements with new sets of devices. In the modified version of the manuscript, we have addressed all the concerns of the reviewer-1. The key changes in the modified manuscript are mentioned below

1. A set of new devices have been fabricated with different area (Supplementary Fig S3c) and constant thickness of AP-ALPO (~130 nm). The output of all such devices has been demonstrated in the Section III, Supplementary Information. It is also demonstrated that the response appears similar in all cases indicating operational scheme at lower device dimensions.
2. The difference in fabrication process, than the conventional process followed with ALPO, is highlighted in the device fabrication section in the manuscript.

Point-by-point reply to the comments of the Reviewer is appended. We hope that the modified version of the manuscript will be acceptable for publication in Nature Communications journal.

Reviewer #1 (Remarks to the Author):

General Comment:

The authors addressed most of the concerns the reviewer raised with extra experiments and analysis. The revised manuscript may be suitable for the publication with additional clarification. One thing still unclear to the reviewer is what the breakthrough of the work is in terms of materials development. ALPO is known as an insulating material and used as a dielectric for low leakage current. The reviewer understands that the trapped charges are controlled by the temperature and thus utilized for a novel non-volatile memory, but the question is whether others simply overlooked this simple temperature-dependent charge control or not. Otherwise, the authors specially treated ALPO for the purpose. This part is not clear. The scalability the reviewer mentioned was the area scalability rather than thickness one. As the authors are aware, the areal dimension should be as small as possible for the high density memory arrays. In fact, some of non-volatile memories lose their memory properties when the capacitor area is reduced to a few tens of nanometers. The reviewer supports the publication after clarification of these issues.

Reply:

We thank the Reviewer for this comment. We agree with the reviewer that the solution processed ALPO is well known and established dielectric for TFT applications. Although we adopted similar chemical growth process of ALPO as reported before, however the memory device has been fabrication followed by oxygen plasma exposure after each processing step. Such fabrication process may help generation of stable traps in ALPO below 200°C. We have highlighted this technique in the manuscript (Section: Device Fabrication). Typically, ALPO based devices are treated at high temperature (>350°C) to achieve better dielectric performance, thus, most of the reported devices do not show any memory effect.

The reviewer has mentioned that some non-volatile memories lose their properties when the area is reduced to nanometers scale. However there is no fundamental reason why this should be applicable to our ALPO made memory devices. This point may be relevant for polymer spin-coated memories where the size of the polymer chain sets a natural length scale for the operation. Our devices are completely inorganic and free from some restrictions. Nevertheless, a set of new devices have been fabricated with different area (Supplementary Fig S3c) and constant thickness of AP-ALPO (~130 nm). The output of all such devices has been demonstrated in the Section III, Supplementary Information. It is also demonstrated that the memory response appears similar in all cases indicating operational scheme at lower device dimension. The diameter of devices is restricted to order of μm which is experimental measurement limit of capacitance.

Reviewer # 2 (Remarks to the Author):

General Comment:

No other comments from my side here.

Reply:

We thank reviewer for accepting our work for publication to Nature Communication. We hope this work will generate further interest in this field.

REVIEWERS' COMMENTS:

Reviewer #1 (Remarks to the Author):

The concerns the reviewers had have been properly addressed in the revised manuscript and thus the manuscript may be suitable for the publication.